# The Impact of Dietary Grape Seed Meal on Healthy and Aflatoxin B1 Afflicted Microbiota of Pigs after Weaning

**DOI:** 10.3390/toxins11010025

**Published:** 2019-01-08

**Authors:** Iulian A. Grosu, Gina C. Pistol, Ionelia Taranu, Daniela E. Marin

**Affiliations:** Laboratory of Animal Biology, National Institute for Research and Development for Biology and Animal Nutrition, Calea Bucuresti no. 1, Balotesti, Ilfov 077015, Romania; grosu.iulian@ibna.ro (I.A.G.); gina.pistol@ibna.ro (G.C.P.); ionelia.taranu@ibna.ro (I.T.)

**Keywords:** microbiota, grape seed, aflatoxin B1

## Abstract

The study investigated the effect of grape seed (GS) meal, aflatoxin (AFB1), or their combination on the large intestine microbiota of weanling piglets. Twenty-four piglets were allocated into four groups based on diet composition: (1) Control group; (2) AFB1 (320 g/kg feed) group; (3) GS group (8% inclusion in the diet); (4) AFB1 + GS group. After 30 days of experiment, the colon content was used for microbiota analyses; after isolation of total bacterial genomic DNA, V3/V4 regions of the 16S rRNA amplicons were sequenced using the Illumina MiSeq platform. The raw sequences were analyzed using the v.1.9.1 QIIME pipeline software. 157 numbers of OTUs were identified among all four dietary groups with 26 of them being prevalent above 0.05% in the total relative abundance. GS and AFB1 increase the relative abundance of phylum *Bacteroidetes* and *Proteobacteria*, while decreasing the *Firmicutes* abundance in a synergic manner as compared with the individual treatments. An additive or synergistic action of the two treatments was identified for *Lactobacillus*, *Prevotella* and *Campylobacter*, while rather an antagonistic effect was observed on *Lachnospira*. The action mechanisms of aflatoxin B1 and grape seed meal that drive the large intestine microbiota to these changes are not known and need further investigations.

## 1. Introduction

Aflatoxins are an important group of mycotoxins mainly produced by the *Aspergillus flavus* and *parasiticus* species, which contaminates a large quantity of the world’s crops, colonizing cereals (especially maize) and other important animal and human food [1,2]. Extensive studies show that aflatoxin B1 is the most harmful of the aflatoxin group, due to its potent toxicity and immune toxicity when is metabolically activated in the liver [2,3]. Chronic exposure to low levels of aflatoxins is a risk factor for hepatocellular carcinoma; based on toxicological data AFB1 has been categorized as a group 1 human carcinogen by the International Agency for Cancer Research [4]. A pig with a diet rich in cereals is very exposed to aflatoxins [5]. Extreme effects can lead to death, but the greatest impact comes from reduced reproductive potential, suppressed immune function, reduced performance, and various pathological effects on organs and tissues [6]. At lower concentrations, AFB1 induced depressed appetite, lower growth rate and lower feed conversion efficiency, decreased proteosynthesis, and immunosuppression especially in younger pigs. An increase in free radicals, leading to unusual oxidative damage and lipid peroxidation, which triggers cell apoptosis was observed [2].

On the other hand, remarkable efforts have been made to develop novel prevention/intervention strategies against AFB1-induced adverse health effects, including liver cancer risks and growth in farm animals. Among the prevention strategies used for the mitigation of AFB1 negative effects, nutritional strategies are the most promising. The nutritional strategies may consist in the use of antioxidants, preservatives and natural inhibitors, activated carbons, dietary fibers, hydrated sodium calcium aluminosilicate, bentonite, zeolites, lactic acid bacteria etc. [7].

Grape seeds are waste products of the winery and grape juice industry [8]. They have recently been used for the production of grapeseed oil which is a valuable food product and a cosmetic ingredient [9]. The seeds contain fatty acids, lipids, amino acids, carbohydrates, and complex fibers, and 4–10% of polyphenols depending on the variety [10]. Grape seed meal is the residual from the grape seeds after oil has been extracted.

Flavonoids represent the predominant class of polyphenols found in grape seed meal, including gallic acid, catechin, epicatechin, gallocatechin, epigallocatechin, and procyanidins [11]. These compounds have been of great interest to the food industry due to their benefits: their anti-aging, anti-inflammation, anti-carcinogenic, anti-mutagenic, anti-ulcer, antiatherogenic, and anti-microbial effects and as inhibitors of human low density lipoprotein oxidation [12]. Recent *in vitro* studies have shown that oxidized polyphenols resulting from tea fermentation can bind AFB1, and that nearly 85% of AFB1 can be transformed into complexed AFB1 [13]. According to the same study, the intestinal absorption of the complexes AFB1-oxidized polyphenols was inhibited in the rat intestine.

Incorporation of industrial by-products in animal diets is an economically and environmentally viable practice for livestock production [14]. The major limitations of using grape seed meal in monogastric feed are the high level of lignified cell wall fraction and the high tannin content. However, *in vivo* and *in vitro* studies carried out over the last few years have shown the beneficial effects of grape seed bioactive compounds administration due to their antioxidant and antimicrobial activity [15].

Oxidative stress caused by AFB1 may be one of the underlying mechanisms for AFB1-induced cell injury [5]. AFB1 enhances ROS formation and causes lipid peroxidation, oxidative DNA and protein damage that can finally lead to tumorigenesis [5]. On the other side, grape seeds have shown important antioxidant properties due to the high polyphenols content [16]. Dietary polyphenols are one of the most important groups of natural antioxidants found in human and animal diets and their antioxidant activity is not only based on directly reacting with ROS but also agonistically activating cellular signaling pathways involved in oxidative stress [17].

Since oxidative stress plays an important role in the toxicity mechanism of AFB1, some antioxidant compounds present in the grape seed meal could be of use in the prevention or at least diminishing the adverse effects of chronic AFB1 toxicity in animals. Some studies have shown that flavonoids as flavone, flavanone and tangeretin can act as an anti-initiators of hepatocarcinogenesis induced by AFB1 through the increase of activity of enzymes involved in the detoxication of AFB1 (glutathione S-transferase, UDP-glucuronyl transferase), increased formation of AFB1-glutathione conjugates and inhibition of the formation of AFB1-DNA adducts. Also, as it was shown in a reconstituted microsomal monooxygenase system, polyhydroxylated flavonoids and phenolic acids can modulate chemical carcinogenesis induced by AFB1 through the inhibition of NADPH-cytochrome P450 reductase [18].

Gut microbiota is a dynamic community of several hundred primarily anaerobic bacteria with important consequences either harmful or advantageous to host physiology and animal performance [19]; these communities live in close proximity to each other developing mainly mutualistic or even symbiotic relationships between them and the host, the composition and abundance of species varying with age, gastrointestinal location, diet, stress, exposure to antibiotics and environmental factors [20]. It participates in the metabolism of the host and take part in the detoxification and metabolic waste excretion [21]. Only approximately 10% of the ingested flavonoid glycosides are absorbed in the upper gastrointestinal tract [22]. Polyphenols cannot exert their beneficial effect in the absence of gut microbiota that can catabolize flavonoids in various kinds of catabolites using a large enzymatic equipment [23].

In recent years, next generation sequencing technologies have revealed all kinds of intricate connections among gut-microbiota, dietary composition and host health [24,25,26]. In this 3-way relationships, oral exposure to xenobiotics or dietary composition could lead to the modulation of gut microbiota, and the changes of gut-microbiota may further influence host health in a significant way [27].

Changing the gut microbiota metabolites, either through changing nutritional habits or by changing the microbiome itself, can also have the potential to mitigate a number of metabolic diseases [17,28]. After our knowledge there are no studies concerning the effect of the concomitant administration of aflatoxin and grape waste on microbiota composition.

The aim of this study was to investigate the AFB1 as a contaminant in feed for its effects at the level of microbiota and whether grape seed meal had the potential to modulate by itself the dynamics of microbiota in pig after weaning and to counteract the possible negative effects on microbiota of pigs fed diet contaminated with AFB1.

## 2. Results

### 2.1. Body Weight and Diarrhea Score

Piglets fed for 30 days with a diet contaminated with AFB1 registered a significantly lower body weight (*p* < 0.05) when compare either with control or GS group (Figure 1). This effect was counteracted by the inclusion of 8% of GS meal into the diet contaminated with AFB1. Reported to the entire period of the experiment only 2.78% of the control piglets had diarrhea. In change, AFB1 was responsible for a higher percentage of piglets with diarrhea (16.67%), while both GS and GS + AFB1 piglets had almost no diarrhea (Figure 2).

### 2.2. 16S rRNA Gene Sequencing and Metagenomic Analysis

The metagenomic analysis of gut microbiota revealed a total of 1,154,427 reads with the lowest sample count of 24,903 and the highest count of 93,622 and an average of 50,192 reads per sample of dietary group, respectively. To remove sampling depth heterogeneity, rarefaction on the OTU table was performed with a cut-off of 24,903, the lowest number of reads recovered in a single sample which was used as the sample depth. More than 90% of sequences from each sample passed quality filtering, showing that the sequencing method is both robust and efficient.

### 2.3. Alpha Diversity

Dietary groups containing Grape Seed Meal, Aflatoxin B1 or both did not result in altered microbial community richness and diversity compared to the control group. The indices inspected included Chao1 (species richness) and PD_whole_tree (phylogenetic diversity units). The lack of influence on richness and diversity was evident at all phylogenetic levels (Figure 3A,B). There were no major community shifts caused by dietary treatments showed in either unweighted (*p* = 0.2019) or weighted (*p* = 0.4817) UniFrac (Figure 4A,B respectively).

### 2.4. Beta Diversity

The Unifrac method, of unweighted and weighted taxon abundances, was used for the construction of the principal coordinate analysis (PCoA) to enable the examination and visualization of the beta diversity within the colon. The Unifrac being phylogeny-based, Weighted-UniFrac takes into account the relative abundance of species/taxa shared between samples. Weighted UniFrac is also useful for examining differences in community structure, which aligned with our study goal. Explaining the low variance of the two axes (22.43% and 10.19%) from the weighted plot can be done considering the multitude of factors that can influence the samples (Figure 5).

### 2.5. Comparisons between Community Compositions

The most highly represented phyla within the large intestine microbiota of the weaning piglets in the control group were *Firmicutes* (58.33%), and *Bacteroidetes* (34.51%), while phyla present in low abundance (<0.01%) included *Proteobacteria* (1.2%), *Spirochaetes*, *Tenericutes*, and *Actinobacteria*, with traces of *Euryachaeota*, *Fibrobacteres*, *Cyanobacteria*, and *Deferribacteres* (Figure 6).

The phylum *Bacteroidetes* was statistically and progressively increased in AFB1 and GS group when compared to control. The *Firmicutes* phylum was also significantly decreased by AFB1 and GS; the concomitant administration of AFB1 + GS have a synergic or additive effect on *Firmicutes* phylum.

In comparison, in the AFB1 group, *Firmicutes* had a 33.2% presence and *Bacteroidetes* 54.4%, in GS group, *Firmicutes* accounted for 32.3% and *Bacteroidetes* 60.3% and in the AFB1 + GS group, *Firmicutes* had just a 12.5% presence while the *Bacteroidetes* reached 73.8%. (Figure 7). The relative abundance of the microbiota at the genus level identified *Prevotella* (21.93%)*, Lactobacillus* (24.8%), *Lachnospira* (0.89%) and *Campylobacter* (0.55%) to be the most abundant in colon. In this study, there were also identified *Megasphaera*, *Anaerovibrio*, *Trembayales*, and *Clostridiaceae* (Figure 8).

As can be seen in the Figure 9, grape seed meal had a stimulating effect on *Prevotella* relative abundance, and GS seems to act synergistic with AFB1, as the highest relative abundance was observed in the AFB1 + GS1 (48.1%) group (*p* < 0.05) when compared to the Control (25.3%) and GS group (39.45%) As shown in Figure 9, both AFB1 and GS significantly impacted (*p* < 0.05) the genus of *Lactobacillus* present in the colon (19% and respectively 17.54%), and they synergistically decrease the *Lactobacillus* abundance from 50.5% in control to 3.8% in the AFB1 + GS dietary group.

The *Lachnospira* genus growth was stimulated significantly (*p* < 0.05) by AFB1 (4.53% vs. 1.69% in control). GS administration doesn’t affect the relative abundance of *Lachnospira* (1.93%) as compared with control. However, a synergic effect on *Lachnospyra* abundance was obtained when AFB1 and GS were administered together. Aflatoxin B1 could also account for a significant growth of 3.03% (*p* < 0.05) of the *Campylobacter* genus in AFB1 group and of 3.8% in AFB1 + GS group while the grape seed meal had no noticeable effect compared to control (0.84% and 1% respectively).

Generally speaking the AFB1 impacted the relative abundance of the large intestine microbiota when compared to the control as seen in the heatmap from Figure 10. AFB1 stimulated some of the bacteria species such as *Campylobacter*, *Lachnospira,* and *Megasphaera*, bacteria from the *Tremblayales* order, the *Paraprevotelaceae* and *Erysipelotrichaceae* microbial families while suppressing bacteria like *Lactobacillus* and *Prevotella*, and some bacteria from the *Lachnospiraceae* family.

On the other hand, the grape seed meal had an almost inverse effect to the AFB1 stimulating the growth of several bacterial species. It positively impacted the growth of *Prevotella, Anaevibrio,* and *Megasphaera, Lachospira*, some bacteria from the order *Clostridiales* and *Tremblayales* while suppressing the rest (Figure 10).

In the GS and AFB1 group the highest value of relative abundance was noted for *Prevotella* and Campylobacter, a modulation of the *Paraprevotelaceae* family and *Bacteroidales* and *Tremblayales* order, while diminishing *Lachnospira* and *Lactobacillus* to their lowest extent.

## 3. Discussions

The present study has demonstrated that the dynamics of the microbiota was changed by grape seed meal, aflatoxin B1 or their combinations and we have identified a complex microbiota in the large intestine of the piglets. This was consistent with the scientific reports which show that large intestine of pig has a high richness and diversity of microbiota [29]. In a comprehensive study published by Kim and collaborators [30] on the pig gastrointestinal microbiome, from a phylum level perspective, most bacteria were classified in five phyla: *Firmicutes*, *Bacteroidetes*, *Proteobacteria*, *Actinobacteria*, and *Spirochaetes*. In this study, bacteria from *Firmicutes* phylum represented the largest proportion of the total population followed by *Bacteroidetes* and these results were similar with our data concerning the distribution of the colon microbiota in piglets. Other studies have shown that these two phyla account for approximately 86–95% of all bacteria present in the large intestine [31].

The most abundant genus identified in the colon of control and experimental piglets from our study were represented by *Prevotella*, *Lactobacillus*, *Lachnospira* and *Campylobacter*. Also, the bioinformatic analysis allowed the identification of *Megasphaera*, *Anaerovibrio*, *Trembayales*, and *Clostridiaceae*.

Most of the bacteria identified in the colon content samples are known to play important roles in the metabolisms of carbohydrates, nucleotides, and amino acids [32]. Our results are similar with other studies that focused on the identification of the core fecal microbiota genus in the colon of the pigs, represented by: *Clostridiales*, *Ruminococcaceae*, *Bacteroidetes*, *Escherichia*, *Firmicutes*, *Lactobacillus*, *Erysipelotrichaceae*, *Lachnospiraceae* [33].

Grape seed meal is rich in polyphenols [34], but only 5–10% of the total polyphenol intake is absorbed in the small intestine, the remaining polyphenols (90–95% of total polyphenol intake) being subject to the enzymatic activities of the gut microbial community [35]. There is evidence from in vitro animal and human studies that phenolic compounds alter gut microbiota and, consequently, alter the *Bacteroides/Firmicutes* balance [36,37]. The *Firmicutes* to *Bacteroidetes* ratio was linked to the general health of vertebrates [38].

In our study, the phylum *Bacteroidetes* was statistically and progressively increased as compared with control group after the addition of GS in the pig diet, while the *Firmicutes* phylum was significantly decreased by GS. Many studies have shown that dietary polyphenols and their metabolites contribute to the maintenance of gut health by the modulation of the gut microbial balance through the stimulation of the growth of beneficial bacteria and the inhibition of pathogen bacteria, exerting prebiotic-like effects [35]. Thus, it was shown that grape seed extracts may modulate the intestinal microbiota, producing changes in beneficial bacteria such as *Lactobacillus* spp. but inhibiting other groups such *Clostridium* spp. in both in vivo and in vitro studies [9,37]. However, in our study, GS has induced a significant decrease of *Lactobacillus*, *Lachnospiraceae*, *Bacteroidales,* and *Campylobacter*, while positively impacted other species like *Megasphera*, *Clostridiales*, *Anaerovibrio,* or *Prevotella.* The higher relative abundance of *Megasphaera*, *Prevotella*, and bacteria belonging to *Clostridiales* order that were found in pigs fed GS diet might have beneficial effects on the host due to the implications of these species in the carbohydrate metabolic chain and the synthesis of short chain fatty acids [39]. *Prevotella*, *Megasphaera*, *Anaerovibrio* represents strictly anaerobic bacteria that have been identified as a dominant species in the large intestine of pigs, and are also abundant in the ileum [29,40,41]. These bacteria are known to be a dominant complex carbohydrate degrading bacterial group in the lower gastrointestinal tract [42] and their increase in the colon of piglet fed a GS diet, might result in response to the higher fiber content of the diet enriched with grape seed meal.

A decrease of *Lactobacillus* abundance as well as an increase in relative abundance of *Clostridium* was already described in a recent study in rats fed with grape pomace extract rich in phenolic compounds [43]. *Lactobacillus* spp. is a beneficial bacterium that enhances gut barrier function, stimulates the host immune system, prevents diarrhea or allergies, contributes to activation of provitamins, and modulates lipid metabolism [44]. An increase or a reduction of growth of *Lactobacillus* was reported in several in vitro and in vivo studies depending on the precursor or the metabolite used. While some phenolic compounds (gallic acid, isoflavones, proanthocyanidins, and resveratrol) increase the relative abundance of *Lactobacillus*, other compounds, (catechin, epicatechin, quercetin) had no significant effect or even decrease the *Lactobacillus* abundance [45]. Studies on pigs fed grape pomace (GP) diets showed that GP decreases the relative abundance of *Lactobacillus* and *Ruminococcus* [46]. Also, the grape seed proanthocyanidins decreased the abundance of *Lactobacillaceae* and increased the abundance of *Clostridiaceae* in both ileal and colonic lumen [47].

Similar with our results, other in vivo studies using grape pomaces have shown an increase of *Clostridiales* abundance. For example, proanthocyanidins induce an increase of *Clostridiales* in pigs fed a diet containing 1% grape seed extract [48] and *Clostridium coccoides* was increased by a wide variety of polyphenolic compounds both in vivo and in vitro [49,50,51].

The *Clostridia* belong to the class of *Firmicutes* and represent obligate anaerobes bacteria, including *Clostridium* and other similar genera [41]. *Clostridium* represent normal constituents of the intestinal flora, with different implication for animal health as some members of the *Clostridiales* family, as *Clostridium difficile* are associated with negative implications, such as inflammatory bowel disease [52], while others as *C. leptum* and *C. coccoides* are important members of the gut flora by promoting healthy aging [53].

Intoxication of pigs with AFB1 was associated with reduced feed intake and body weight gain, impaired liver and immune functions and altered serum biochemical parameters [54]. Indeed, our results have shown a significant decrease in the body weight of AFB1 intoxicated piglets. Even that diarrhea was less associated with aflatoxin B1 intoxication when compared with other mycotoxin as deoxynivalenol [55], our study has shown an increase of the number of pigs with diarrhea after the AFB1 administration as compared with the control. Also, in our experiment, intoxication with AFB1 induced compositional changes in microbial community, consisting in an increase of the phylum *Bacteroidetes*, while decreasing the *Firmicutes* phylum abundance, but the changes were opposites when compared with the changes induced by GS diet. Similar results were obtained also in rats exposed to different concentration of AFB1, species whose compositions significantly decreased among the top 20 OTUs belong mainly to *Firmicutes*, with an important decrease of *Lactobacillus*, mainly of *Streptococcus* sp. and *Lactococcus* sp. [1]. When looking to the species affected by the AFB1 exposure, the toxin induced a decrease of *Prevotella* and *Lactobacillus* in the colon of intoxicated piglets, while increasing *Camplyobacter*, *Lachnospira* and some members of *Erysipelotrichaceae*, *Bacteroidales* or *Tremblayales* families. Earlier studies have shown that the antimicrobial spectrum of aflatoxin B1 is narrow and limited and AFB1 was found to be inactive against common Gram-positive and Gram-negative bacteria at a concentration of 100 µg/mL [56]. Thus, a general antimicrobial effect cannot explain the alterations of microbiota found in the colon of intoxicated animals. It rather looks that other more complex mechanisms can be involved and it was suggested that the gut microbiota of a host could be intentionally modified to accumulate toxin-tolerant species, a potential natural biological intervention strategy for mitigation of AFB1-induced toxic effect [1]. For example, it has been reported that some lactic acid bacteria can remove AFB1 or have protective effects against AFB1 [57,58]. Indeed, an increase of AFB1 concentration in feces was observed in intoxicated mice receiving *L. plantarum* as compared with mice receiving only AFB1 [59].

Less piglets with diarrhea were observed in the group fed AFB1 + GS diet as compared with AFB1 group, suggesting that GS was responsible for a shift in the microbiota that results in a reduction of diarrhea. Concomitant administration of GS and AFB1 caused significant shift in the microbial community by significantly increasing the relative abundance of phylum *Bacteroidetes* and *Proteobacteria*, while decreasing the *Firmicutes* abundance in a synergic or additive manner as compared with the individual treatments. Again, an additive or synergistic action of the two treatments was identified for *Lactobacillus*, *Prevotella*, and *Campylobacter*, while rather an antagonistic effect was observed on *Lachnospira*. *Lachnospira* decrease observed in the colon of piglets fed both AFB1 and GS can be correlated with the reduction of diarrhea incidence as compared with AFB1 group, as *Lachnospira* is positively correlated with diarrhea in piglets [60]. Also, it was shown that the abundance of *Prevotella* was increased in healthy pigs compared with pigs with diarrhea and the higher abundance of *Prevotellaceae* family in healthy pigs may provide an adequate prevention strategy for pathogen infection [61]. However, our microbiota results cannot entirely explain the beneficial effect of grape seed meal, and other microbiota-independent mechanisms might be responsible for the positive effects of the grapeseed meal in weaned piglets. Indeed, our preliminary data have shown that the inclusion of grape seed in the diet of AFB1 intoxicated pigs restored toward the control group the level of the phase-II antioxidant enzymes activity and total antioxidant capacity, decreased TBARS level, and had the potential to ameliorate the pro-inflammatory cytokines concentration and the performance of AFB1-treated animals (unpublished data).

In conclusion, piglets fed for 30 days with a diet contaminated with AFB1 registered a significantly lower body weight and a high frequency of diarrhea. These effects were counteracted at least partially by the inclusion of 8% GS meal into the diet contaminated with AFB1. In the analyses concerning large intestine microbiota 157 numbers of OTUs were identified among all four dietary groups with 26 of them being prevalent above 0.05% in the total relative abundance. Most of the OTUs were identified as bacteria involved in polysaccharides (hemicellulose and pectin, host glycan, and α-glucan) and carboxylic and amino acid metabolism. Concomitant administration of GS and AFB1 caused significant shift in the microbial community by significantly increasing the relative abundance of phylum *Bacteroidetes* and *Proteobacteria*, while decreasing the *Firmicutes* abundance in a synergic or additive manner as compared with the individual treatments. Again, an additive or synergistic action of the two treatments was identified for *Lactobacillus*, *Prevotella* and *Campylobacter*, while rather an antagonistic effect was observed on and *Lachnospira.* The action mechanisms of aflatoxin B1 and grape seed meal that drive the large intestine microbiota to these changes are not known and need further investigations.

## 4. Materials and Methods

### 4.1. Animals and Diets

Animals were handled in accordance with the Romanian Law 206/2004 and the EU Council Directive 98/58/EC for handling and protection of animals used for experimental purposes. The study protocol was approved by the Ethical Committee of the National Research-Development Institute for Animal Nutrition and Biology, Balotesti, Romania (Ethical Committee no. 52/2014, Date of approval: 10 March 2014).

Twenty-four healthy weaned crossbred TOPIGS-40 hybrid piglets, with an average body weight of 9.13 ± 0.03 kg were allocated into four groups of 6 piglets based on diet composition as follows: (1) Control group fed a standard diet; (2) AFB1 group fed a diet contaminated with 320 µg/kg AFB1; (3) GS group fed a diet supplemented with a 8% grape seed meal; (4) AFB1 + GS group fed a diet containing both 320 µg/kg AFB1 and 8% grape seed meal. The grape seed meal provided by a local commercial (S.C. OLEOMET-SA S.R.L., Bucuresti, Romania) was included in the diet by corn replacement. The total polyphenol content and identification of different classes of polyphenols of grape seed meal was measured by the Folin–Ciocalteu reaction and HPLC-DAD-MS and gas chromatography as described by [62]. The polyphenols composition of the grape seed meal is presented in Table 1.

To prepare the toxin contaminated diet, 50 mg AFB1 (FERMENTEC, Jerusalem, Israel) was dissolved in DMSO (dymethil sulfoxide) and mixed into basal diet to provide a feed diet containing 320 μg/kg; final AFB1 concentration in feed was verified by ELISA using a Veratox kit (Neogen, Lansing, MI, USA). The content of other mycotoxins that frequently contaminate cereals (DON, ZEA, OTA) was under the detection limit as resulted from the ELISA assay.

The animals were individually identified by ear tag and housed in pens (two replicates of 3 pigs per pen per treatment) and fed the experimental diets for 30 days. They had free access to feed and water every day of the experimental period. The piglets with diarrhea were recorded every day of the experiment. All diets were formulated to meet specific requirements for weaning feed as indicated by NRC (2012)—Table 2. At the end of the experiment pigs were weighted and then slaughtered. The content of the colon was immediately frozen and stored at −80 °C until use.

### 4.2. Microbial DNA Extraction and 16S rRNA Gene Sequencing

Microbial DNA was extracted using the QIAamp DNA stool minikit (QIAGEN, Dusseldorf, Germany) following the manufacturer protocol additionally increasing the temperature to 95 °C to improve the DNA yield. Sample DNA concentrations were verified on agarose gel electrophoresis and on the Nanodrop Spectrophotometer and adjusted to 10 ng/μL with a volume of 100 μL for each sample for a total of 24 samples.

Microbial profiling was performed by BMR Genomics (Padova, Italy) using the Illumina Miseq platform using 300PE approach on 16S amplicons derived from the bacterial DNA. The library formation and sequencing of the 16S rRNA genes were carried out according to the method above. In the first step the V3 and V4 regions of the amplicons were amplified with universal primers: forward primer: 5′-CCTACGGGAGGCAGCAGT-3′ and reverse primer 5′-GACTACCAGGGTATCTAATCCTGTT-3′ [63] and forward primer: 5′-CCTACGGGNBGCASCAG-3′ and reverse primer: 5′-GACTACNVGGGTATCTAATCC-3′ [64]. In the second amplification step, the adaptors and indexes were bound to the sequences. The pooling of the samples was followed by a cluster generation and sequencing with the Paired End 2 × 300 pb format.

### 4.3. Microbiota Bioinformatics

For the microbial raw data, the sequences obtained were investigated using a subsampled open reference OTU (operational taxonomic unit) picking method with default settings performed in QIIME (v1.9.1). The 16S rRNA gene reads were then paired-end and demultiplexed. Subsampled open-reference analyzable samples were clustered into Operational Taxonomic Units. The OTU-picking strategy was carried out by using UCLUST algorithm using a de-novo strategy with a 97% sequence similarity threshold. Representative sequences for each OTU were set in a matchup with the Greengenes database V13_8 and taxonomy was designated through the application of the UCLUST method with a 90% confidence threshold. Data was chimera checked using the Blast fragments approach in QIIME. The raw reads in QIIME were filtered for a length ≥70 and an average quality ≥20. In order to eliminate sampling depth heterogeneity, the OTU table was rarefied to the lowest number of reads obtained from a single sample. Furthermore, the OTU table obtained in the previous step was abundance filtered by removing the OTUs with a relative abundance ≤0.005% across all samples in order to keep only the most abundant species for further data processing.

A phylogenetic tree was constructed by firstly aligning the representative sequences of each OTU using PyNAST algorithm against Greengene reference sequences. Above mentioned procedures were implemented through QIIME with customized commands. The OTU table and phylogenetic tree were the major elements for data analysis.

The diversity was estimated with Chao 1 and Faith’s phylogenetic diversity (PD_whole_tree) observed-species indices (alpha or within-sample diversity) and double principal component analysis (DPCoA; beta or between-sample diversity) using the phylogeny-based weighted Unifrac distance matrix.

A heatmap was also built around the OTU table of the species that were found above a 0.005% relative abundance.

### 4.4. Statistical Analysis

Analyses were performed using XLstat software package (http://www.xlstat.com). Comparison between groups was performed using one-way analysis of variance (ANOVA) with the post hoc Tukey test for multiple comparisons at the *p* < 0.05 level. ANOSIM is an internal statistical method (script) of the QIIME which was used to test the differences between two or more samples. The ANOSIM test is nonparametric and the statistical significance is derived through permutations.

## Figures and Tables

**Figure 1 toxins-11-00025-f001:**
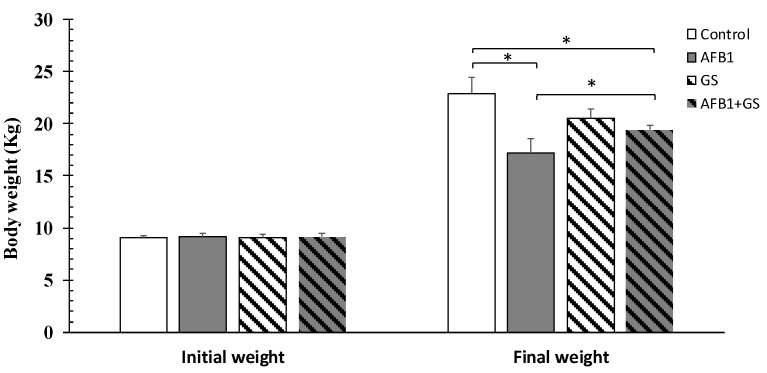
Effect of different dietary treatments of piglets’ body weight. GS: grape seed; AFB1: aflatoxin B1. * indicate significant differences (*p* < 0.05) between groups.

**Figure 2 toxins-11-00025-f002:**
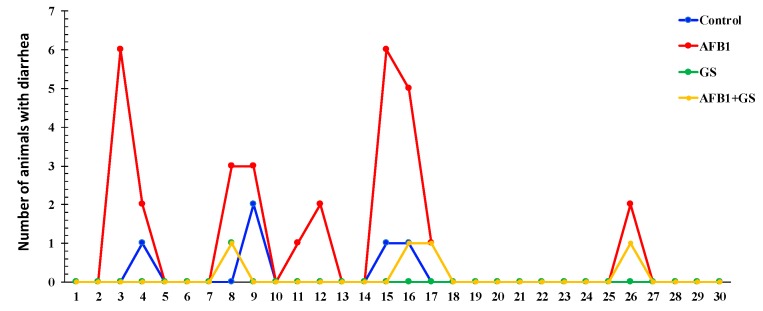
Effect of different dietary treatments on the number of piglets with diarrhea. GS: grape seed; AFB1: aflatoxin B1.

**Figure 3 toxins-11-00025-f003:**
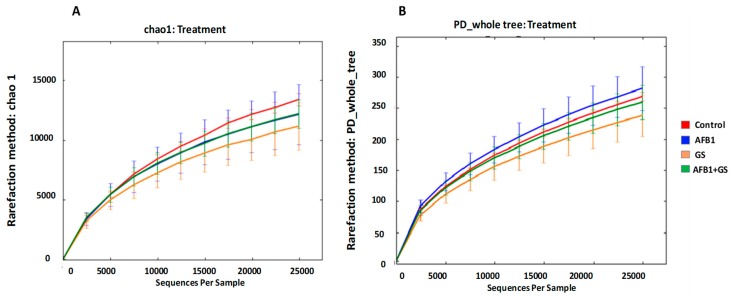
Analysis of alpha diversity in gut microbiota of weaning piglets fed with control diet, AFB1-treated, GS diet and GS + aflatoxin B1 (AFB1) diet. The diversity indices for each experimental group was predicted by Chao 1 (**A**) or by phylogenetic diversity (**B**). GS: grape seed; AFB1: aflatoxin B1.

**Figure 4 toxins-11-00025-f004:**
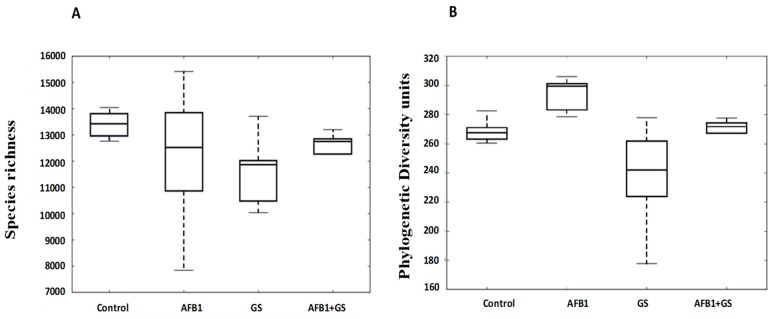
Alpha diversity box plots of Chao 1 (**A**) and of phylogenetic diversity (**B**) showing richness profiles for each experimental diet. GS: grape seed; AFB1: aflatoxin B1.

**Figure 5 toxins-11-00025-f005:**
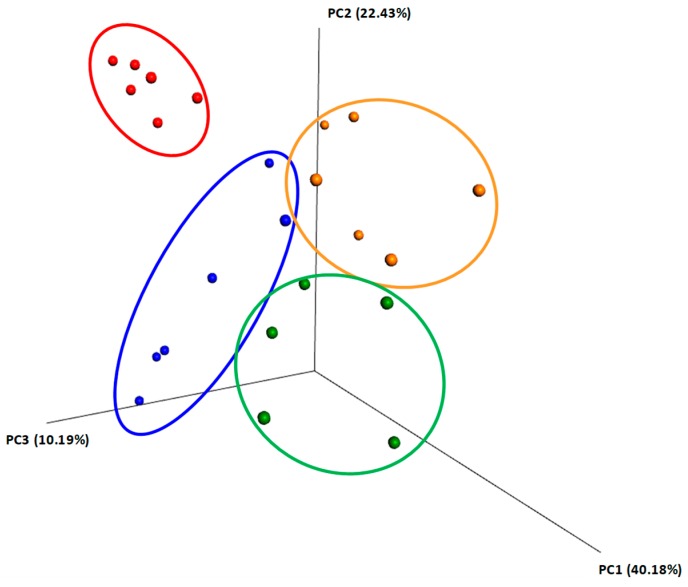
Principal component analysis plot based on OTUs found in the content of colon collected from weaning piglets fed different diets. Plot is based on unweighted Unifrac Distances. The amount of variance is depicted by the percentages in parentheses on each axis. Ellipses denote clustering according to each type of experimental diets: control (red), AFB1 (blue), GS (orange) and GS + AFB1 (green). GS: grape seed; AFB1: aflatoxin B1.

**Figure 6 toxins-11-00025-f006:**
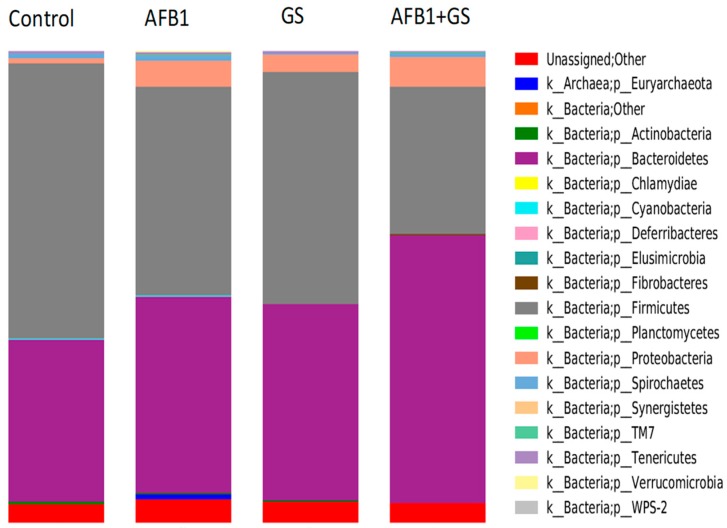
Relative abundances (%) of bacteria phyla found in the content of colon collected from weaning piglets fed different experimental diets: control, AFB1, GS, and GS + AFB1. GS: grape seed; AFB1: aflatoxin B1.

**Figure 7 toxins-11-00025-f007:**
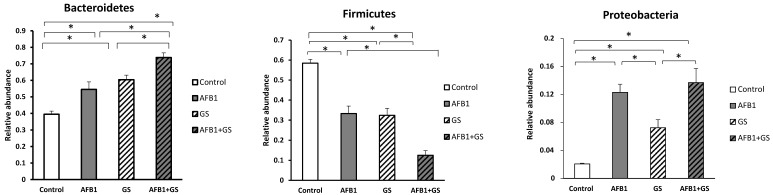
The relative abundances of the most important phyla found in the content of colon collected from weaning piglets. GS: grape seed; AFB1: aflatoxin B1. * indicate significant differences (*p* < 0.05) between groups.

**Figure 8 toxins-11-00025-f008:**
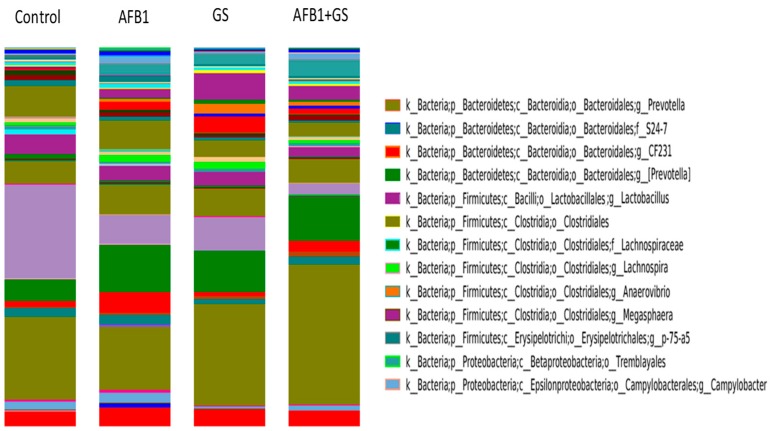
Relative abundances (%) of bacteria genus found in the content of colon collected from weaning piglets fed different experimental diets: control, AFB1, GS, and GS + AFB1. GS: grape seed; AFB1: aflatoxin B1.

**Figure 9 toxins-11-00025-f009:**
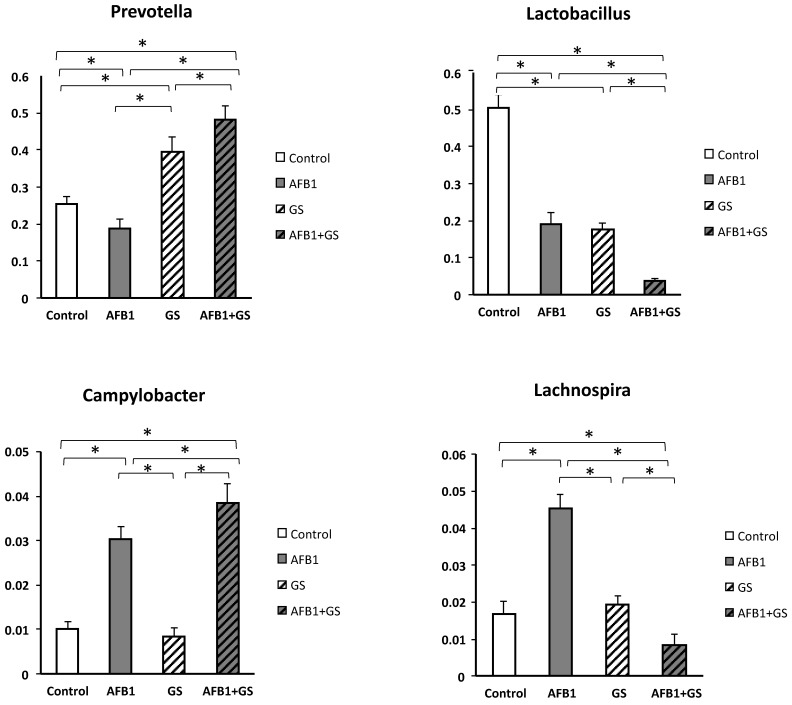
The relative abundances of the most important genus found in the content of colon of weanling piglets. GS: grape seed; AFB1: aflatoxin B1. * indicate significant differences (*p* < 0.05) between groups.

**Figure 10 toxins-11-00025-f010:**
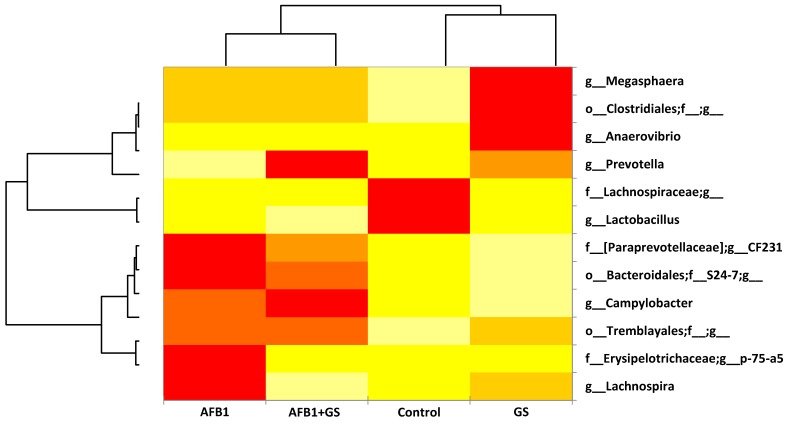
Heatmap illustrating the most important genus found in the content of the colon of weanling piglets. GS: grape seed; AFB1: aflatoxin B1.

**Table 1 toxins-11-00025-t001:** The composition in polyphenols of the grape seed meal.

Polyphenols	Grape Seed Meal
Total polyphenols (mg GAE eq/100 g)	5567.2
Phenolic compound (mg Quercitin eq/100 g)	
Ferulic acid derivative	24.10
Caffeoilquinic acid	40.15
Procianidin trimer	9.11
Catechin	9.53
Epicatechin	8.51
Gallocatechin	8.60
Epigalocatechin	15.60
Procianidin dimer	11.09
Petunidin 3-O-glucoside	12.15
Procianidin dimer	10.08
Ferulic acid	12.43
Cianidin coumaroil-glucoside	12.79
Malvidin coumaroil-glucoside	14.26
Dicaffeoilquinic acid	19.50
Dicaffeoilquinic acid	20.20

**Table 2 toxins-11-00025-t002:** Composition of experimental diet (%).

Ingredients (%)	Control	AFB1	GS	GS + AFB1
Corn	67.47	67.47	58.5	58.5
Soybean meal	19	19	18	18
Gluten	4	4	4	4
Milk replacer	5	5	5	5
Soya oil	-	-	2	2
L Lysine	0.4	0.4	0.4	0.4
DL Methionine	0.1	0.1	0.15	0.15
Monocalcium phosphate	1.46	1.46	1.33	1.33
Feed grade limestone	1.37	1.37	1.42	1.42
Salt	0.1	0.1	0.1	0.1
Choline premix	0.1	0.1	0.1	0.1
Vitamin mineral premix ^1^	1	1	1.0	1.0
Grape seed meal	-	-	8	8
AFB1 (mg/kg)	-	320	-	320
**Analyzed composition**				
EM (kcal/kg)	3248	3248	3178	3178
Crude protein %	18.42	18.42	18.21	18.21
Lisine %	1.2	1.2	1.2	1.2
Methionine +Cysteine%	0.72	0.72	0.72	0.72
Ca%	0.9	0.9	0.9	0.9
P%	0.65	0.65	0.65	0.65
Fat%	3.03	3.03	3.19	3.19
Celulose%	3.12	3.12	5.8	5.8

^1^ Vitamin-mineral premix/kg diet: (0–18 days): 10,000 UI vit. A; 2000 vit. D; 30 UI vit. E; 2 mg vit. K; 1.96 mg vit. B_1_; 3.84 mg vit. B_2_; 14.85 mg pantothenic ac.; 19.2 mg nicotinic ac.; 2.94 mg vit. B_6_; 0.98 mg folic ac.; 0.03 mg vit. B_12_; 0.06 biotin; 24.5 mg vit. C; 40.3 mg Mn; 100 mg Fe; 100 mg Cu; 100 mg Zn; 0.38 I; 0.23 mg Se.

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
