# Peer review of "The Impact of Dietary Grape Seed Meal on Healthy and Aflatoxin B1 Afflicted Microbiota of Pigs after Weaning"

_toxins, 2019, doi:10.3390/toxins11010025_

Round 1
Reviewer 1 Report
The manuscript ID : toxins-403729, entitled: « The impact of dietary grape seed meal on healthy and aflatoxin afflicted microbiota of pigs after weaning » have highlighted the effects of dietary of grape seed (GS) meal, aflatoxin (AFB1) or their combination on the large intestine microbiota of weanling piglets.
1/Minor comments about the present manuscript :
ü Introduction: references need to be completed (lanes 78-85)
ü Lane 102 : 16S rRNA Gene sequencing and data preparation : metagenomic analysis
ü Lane 119 : Fig. 3A and 3B : number to be changed
ü Lane 121 : Fig. 4A and 4B : number to be changed
ü Lane 150 : Megasphaera, Anaerovibrio, Trembayales, and Clostridiaceae : in italic
ü Lane 152 : Legend is missing and need to be completed. I propose figure 6 A and Fig. 6B
ü Lane 155 : control (red), AFB1 (blue), GSM (orange) and GSM+AFB1 (green). : colour rectangles should be removed as their colours can be confused with the colours used for bacterial species
ü Lane 161 : Relative abundances (%) is missing in the fig. 8
ü Lane 224 : ( [9,26].
ü Lane 230 : Prevotella : police to be changed
ü Lane 312 : units are missing
ü Lane 335 : 95°C
ü Lane 341 : 16 S rRNA
ü Table 2 : the meaning of genes must be added
ü Lanes 379, 384, 483, 486 : Aflatoxin B1
ü Lanes 380, 385 : F344 rats
ü Lane 484 : lactobacillus brevis and lactobacillus paracasei : in italic
ü Lane 486 : lactobacillus and 486 propionibacterium: in italic and space propioni bacterium:
2/Major comments :
ü Figures should be controlled and clarified (ex Fig 6 A/B)
ü in the discussion part lane 276 : « Thus, the decrease of Lactobacillus observed in our study can be due to their ability to bind AFB1, to reduce the toxin absorption at intestinal level and to eliminate the complex AF-bacteria through feces » : this point must be demonstrated and complemented by an experimental approach.
ü In order to correlate the decrease/increase of a bacterial species with dietary grape seed meal, a biochemical analysis of GS would be appreciated and could be added to the document.
Author Response
Reviewer 1
Comments and Suggestions for Authors
The manuscript ID: toxins-403729, entitled: « The impact of dietary grape seed meal on healthy and aflatoxin afflicted microbiota of pigs after weaning » have highlighted the effects of dietary of grape seed (GS) meal, aflatoxin (AFB1) or their combination on the large intestine microbiota of weanling piglets.
1/Minor comments about the present manuscript :
ü Introduction: references need to be completed (lanes 78-85)
ü Lane 102 : 16S rRNA Gene sequencing and data preparation : metagenomic analysis
ü Lane 119 : Fig. 3A and 3B : number to be changed
ü Lane 121 : Fig. 4A and 4B : number to be changed
ü Lane 150 : Megasphaera, Anaerovibrio, Trembayales, and Clostridiaceae : in italic
ü Lane 152 : Legend is missing and need to be completed. I propose figure 6 A and Fig. 6B
ü Lane 155 : control (red), AFB1 (blue), GSM (orange) and GSM+AFB1 (green). : colour rectangles should be removed as their colours can be confused with the colours used for bacterial species
ü Lane 161 : Relative abundances (%) is missing in the fig. 8
ü Lane 224 : ( [9,26].
ü Lane 230 : Prevotella : police to be changed
ü Lane 312 : units are missing
ü Lane 335 : 95°C
ü Lane 341 : 16 S rRNA
ü Lanes 379, 384, 483, 486 : Aflatoxin B1
ü Lanes 380, 385 : F344 rats
ü Lane 484 : lactobacillus brevis and lactobacillus paracasei : in italic
ü Lane 486 : lactobacillus and 486 propionibacterium: in italic and space propioni bacterium:
Author response: All the modifications suggested by the reviewer (lines 102-486) can be found now in the new version of the manuscript.
Table 2 : the meaning of genes must be added
We agree with the Reviewer 1, that the table 2 “Universal primers used in amplification of the V3 and V4 regions” and the explanation of this table was not clear enough. Microbial profiling was performed by BMR Genomics (Italy) using the Illumina Miseq platform using 300PE approach on 16S amplicons derived from the bacterial DNA. The 16S V3 and V4 regions of the amplicons were amplified with universal primers for Bacteria and Archaea: forward primer: 5′-CCTACGGGAGGCAGCAGT-3′ and reverse primer 5′-GACTACCAGGGTATCTAATCCTGTT-3′ (Nadkarni et al.,2002) and forward primer: 5′-CCTACGGGNBGCASCAG -3′ and reverse primer: 5′-GACTACNVGGGTATCTAATCC -3′ (Takahashi et al. 2014). Consequently, we have chosen to delete table 2 and to insert instead the relevant information about primers in the new version of M&M:
“In the first step the V3 and V4 regions of the amplicons were amplified with universal primers: forward primer: 5′-CCTACGGGAGGCAGCAGT-3′ and reverse primer 5′-GACTACCAGGGTATCTAATCCTGTT-3′ [63] and forward primer: 5′-CCTACGGGNBGCASCAG -3′ and reverse primer: 5′-GACTACNVGGGTATCTAATCC -3′ [64].”
2/Major comments:
ü Figures should be controlled and clarified (ex Fig 6 A/B).
Author response: The figures were checked and clarified.
ü in the discussion part lane 276 : « Thus, the decrease of Lactobacillus observed in our study can be due to their ability to bind AFB1, to reduce the toxin absorption at intestinal level and to eliminate the complex AF-bacteria through feces » : this point must be demonstrated and complemented by an experimental approach.
Author response: the reviewer has wright and the authors are very grateful for this observation. Indeed, we didn’t demonstrate yet that the decrease of Lactobacillus is due to their ability to bind AFB1 and subsequently the excretion of the complexes AFB1-bacteria through feces. This was only a supposition as many studies have shown that Lactobacillus have the ability to bind AFB1 as shown in the discussion section: ”For example, it has been reported that some lactic acid bacteria can remove AFB1 or have protective effects against AFB1 (Goma et al., Huang et al., 2017). But, being only a supposition the phrase ”Thus, the decrease of Lactobacillus observed in our study can be due to their ability to bind AFB1, to reduce the toxin absorption at intestinal level and to eliminate the complex AF-bacteria through feces” was removed from the discussion. This paper intended to analyze the effects of dietary grape seed meal, aflatoxin B1 or their combination on the large intestine microbiota of weanling piglets and we didn’t perform any analyses of mycotoxines in the faeces as in this moment our laboratory has a limited expertise in mycotoxin analyse using HPLC (we have just started to identify and characterize mycotoxins from complex samples (feed, blood, tissues, faeces etc). So for the moment we cannot perform an additional experiment in order to demonstrate our hypothesis.
Author response:
ü In order to correlate the decrease/increase of a bacterial species with dietary grape seed meal, a biochemical analysis of GS would be appreciated and could be added to the document.
Author response:
As suggested, the composition in polyphenols was added in Table 1 in the new version of the manuscript. All details concerning the polyphenols assessment, as well as a table (Table 1) with total polyphenols composition, as well as the different classes of polyphenols identified in the grape seed meal were included into the M&M section.
Reviewer 2 Report
Dear authors,
After the review proces I have several comments:
- the article has an important and interesting aim, if the quantity of grape seed could be obtained with minimum impact in farm management.
- you should discuss the influence of bioactive compounds in aflatoxin inactivation.
- you should detail the oxidative stress mechanisms of aflatoxin and antioxidant potential of grape seed meal.
- you should present which flavonoidic compounds (Page 2, Line 57-61) are the most important for aflatoxin stress combat.
- you should detail how the microbiota could influence the presence and level of flavonoids.
Best regards!
Author Response
- the article has an important and interesting aim, if the quantity of grape seed could be obtained with minimum impact in farm management.
Author response: Grape seed (GS) represents a waste resulted from oil extraction and it is available in quite large quantities as a dried by-product. GS meal is rich in bioactive compounds which mitigate the aflatoxin B1 negative effects (Taranu et al., under revision in Toxicon) and in addition the administration of GS diets in farm animals increase the nutritional value of meat (Kafantaris et al., 2018; Nardoia et al., 2017 etc ). In the general context of the increasing consumer demands for healthier and qualitative animal products, farmers are interested to use new feed ingredients, as grape seed waste that can contribute to an increase nutritional value of animal products.
- you should discuss the influence of bioactive compounds in aflatoxin inactivation.
Author response: Thank you for this interesting suggestion. Indeed, recent studies have shown that AFB1 can be complexed by polyphenols from tea and that the intestinal absorption of complexes AFB1-polyphenols is inhibited in rats (Lu et al., 2017). According to the reviewer suggestion, we have included these findings in the Introduction section:
“Recent in vitro studies have shown that oxidized polyphenols resulted from tea fermentation can bind AFB1, and nearly 85% of AFB1 can be transformed into complexed AFB1 [13]. According to the same study, the intestinal absorption of the complexes AFB1-oxidized polyphenols was inhibited in the rat intestine.”
- you should detail the oxidative stress mechanisms of aflatoxin and antioxidant potential of grape seed meal.
Author response: We agree with the reviewer that the effect of both AFB1 and grape seed meal on oxidative stress were not explained. According to the reviewer suggestion, we have included these information into the new version of the paper:
“Oxidative stress caused by AFB1 may be one of the underlining mechanisms for AFB1-induced cell injury [5]. AFB1 enhances ROS formation and causes lipid peroxidation, oxidative DNA and protein damage that can finally lead to tumorigenesis [5]. On the other side, grape seeds have shown important antioxidant properties due to the high polyphenols content [16]. Dietary polyphenols are one of the most important groups of natural antioxidants found in human and animal diets and their antioxidant activity is not only based on directly reacting with ROS but also agonistically activating cellular signaling pathways involved in oxidative stress [17].”
- you should present which flavonoidic compounds (Page 2, Line 57-61) are the most important for aflatoxin stress combat.
Author response: According to the reviewer suggestion, some examples of flavonoidic compounds that can counteract AFB1 negative effect were included into the new version of the manuscript.
“Some studies have shown that flavonoids as flavone, flavanone and tangeretin can act as an anti-initiator of hepatocarcinogenesis induced by AFB1 through the increase of activity of enzymes involved in the detoxication of AFB1 (glutathione S-transferase, UDP-glucuronyl transferase), increase of the formation of AFB1-glutathione conjugates and inhibition of the formation of AFB1-DNA adducts. Also, as it was shown in a reconstituted microsomal monooxygenase system, polyhydroxylated flavonoids and phenolic acids can modulate chemical carcinogenesis induced by AFB1 through the inhibition of NADPH-cytochrome P450 reductase [18].”
- you should detail how the microbiota could influence the presence and level of flavonoids.
Author response: Thank you for this suggestion. Below we explained microbiota influence
Gut microbiota affects the bioavailability and effects of polyphenols through enzymatic transformation such as dehydroxylation, decarboxylation, demethylation and other processes [1]. Bacterial glycohydrolases modify the bioavailability of glycosides by transforming them into aglycones, active isoflavones with an activity similar to estrogens are produced via bacteria in the microbiota, similarly quercitin generated by the microbial enzymes have a down-regulating effect on the inflammatory responses [2,3].
Microbial metabolites resulted from phenolics and flavonoids can also show antimicrobial or bacteriostatic activities, influencing further the microbiota composition [4]. Polyphenols and flavonoids can also modify other properties of bacteria such as reduced adhesion ability, production of mucin and modulation of bacterial intestine colonization [5,6]. More studies need to be done to elucidate other interactions between polyphenols and gut microbiota functions although the bioavailability and the extensive range of flavonoid microbial metabolites that are formed after ingestion that we know of supports a strong relation between flavonoids and microbiome [7].
As suggested we have included in the Introduction section an explanation concerning the role of microbiota in the metabolism of flavonoids as follow:
“Only approximately 10% of the flavonoid glycosides ingested are absorbed in the upper gastrointestinal tract [22]. The polyphenols cannot exert their beneficial effect in the absence of gut microbiota that can catabolize flavonoids in various kinds of catabolites using a large enzymatic equipment [23]”.
Reviewer 3 Report
In this manuscript, the authors show that grape seed meal in the diet of pigs significantly improved weight gain and diarrhoea responses to challenge with aflatoxins. The work is interesting as the use of natural feed additives to promote health in livestock is a very topical area. However, some aspects of the manuscript must be improved.
1) The overall level of English is not adequate - it must be revised by a native speaker or a professional editing service.
2) The composition of the grape seed meal should be reported. Particularly the total polyphenol content, and if possible the composition of the polyphenols (e.g. % of flavanols, procyanidins etc.). This is crucial for comparison to other studies.
3) The hypothesis for the study was that grape seed meal would improve pig performance by alleviating aflatoxin-induced changes in the microbiota. Whilst the data would suggest that pig health was improved, the extent to which this was due to effects on the microbiota is not clear – in most cases, the grape meal and the toxin seem to have similar effects (and indeed additive) effects on the microbiota (epscially Lactobacillus). Whilst an increase in prevotella may indeed be connected to improved gut health, overall it is not clear if the results support the hypothesis – and if other, microbiota-independent mechanisms may be responsible for the positive effects of the grape meal. This is not gone into detail in the discussion.
4) The figures and the legends need some work – especially Figure 6 seems to present both phylum and genus-level data but the legend refers only to phylum.
5) Finally, a number of recent relevant manuscripts which have investigated the effects of grape meal on gut health in pigs are not cited here: e.g. Williams et al. 2017 (PLoS ONE e0186546) and Han et al. 2016 (Oncotarget 7:49) both report a similar reduction in Lactobacillus in pigs fed grape meal. These would seem to be quite relevant – perhaps also the work of Fiesel et al. 2014 (BMC Veterinary Research 10:196).
Author Response
Comments and Suggestions for Authors
In this manuscript, the authors show that grape seed meal in the diet of pigs significantly improved weight gain and diarrhoea responses to challenge with aflatoxins. The work is interesting as the use of natural feed additives to promote health in livestock is a very topical area. However, some aspects of the manuscript must be improved.
1) The overall level of English is not adequate - it must be revised by a native speaker or a professional editing service.
Author response: The manuscript was revised by Dr. Catarina Gosh (PAREXEL International Corporation, United Kingdom and by Mr. Mihai Roman (English Translator, INCDBNA-IBNA, Balotesti, Romania).We have added their contribution into the Acknowledgement section.
“The authors thanks to Dr. Catarina Gosh and to Mr. Mihai Roman for proof reading this manuscript”
2) The composition of the grape seed meal should be reported. Particularly the total polyphenol content, and if possible the composition of the polyphenols (e.g. % of flavanols, procyanidins etc.). This is crucial for comparison to other studies.
Author response: As suggested, the composition in polyphenols was added in the new version of the manuscript (Table 1). All details concerning the polyphenols assessment, as well as a table (Table 1) with total polyphenols composition, and the different classes of polyphenols identified in the grape seed meal were included into the M&M section.
3) The hypothesis for the study was that grape seed meal would improve pig performance by alleviating aflatoxin-induced changes in the microbiota. Whilst the data would suggest that pig health was improved, the extent to which this was due to effects on the microbiota is not clear – in most cases, the grape meal and the toxin seem to have similar effects (and indeed additive) effects on the microbiota (especially Lactobacillus). Whilst an increase in Prevotella may indeed be connected to improved gut health, overall it is not clear if the results support the hypothesis – and if other, microbiota-independent mechanisms may be responsible for the positive effects of the grape meal. This is not gone into detail in the discussion.
Author response: We really appreciate your comment. Indeed, the microbiota data does not support entirely the benefic effect of grape seed polyphenols in counteracting the negative effects of AFB1. The results presented in this paper represent a part from a large experiment carried out in our laboratory in which other aspects of the interaction AFB1-grape seed meal such as effect on inflammation, oxidative stress at different tissue levels were investigated obtained. Our results have shown that the inclusion of grape seed in the diet of AFB1 intoxicated pigs restored toward the control group the level of phase-II antioxidant enzymes activity and total antioxidant capacity, decreased TBARS level, and had the potential to ameliorate the pro-inflammatory cytokines concentration and the performance of AFB1-treated animals also toward that of control group (Taranu et al., under revision in Toxicon). Thus, it is more than probable that others, microbiota-independent mechanisms, might be responsible for the positive effects of the grape meal. This aspect is now commented/explained in the new version of the paper as follow:
“However, our microbiota results cannot entirely explain the beneficial effect of grape seed meal, and other mechanisms, microbiota-independent might be responsible for the positive effects of the grapeseed meal in weaned piglets. Indeed, our preliminary data have shown that the inclusion of grape seed in the diet of AFB1 intoxicated pigs restored toward the control group the level of the phase-II antioxidant enzymes activity and total antioxidant capacity, decreased TBARS level, and had the potential to ameliorate the pro-inflammatory cytokines concentration and the performance of AFB1-treated animals (unpublished data).”
4) The figures and the legends need some work – especially Figure 6 seems to present both phylum and genus-level data but the legend refers only to phylum.
Author response: The reviewer has wright. It was a mistake in the Figure numbers. All the Figures and their corresponding legends were revised and corrected in the new version of the paper.
5) Finally, a number of recent relevant manuscripts which have investigated the effects of grape meal on gut health in pigs are not cited here: e.g. Williams et al. 2017 (PLoS ONE e0186546) and Han et al. 2016 (Oncotarget 7:49) both report a similar reduction in Lactobacillus in pigs fed grape meal. These would seem to be quite relevant – perhaps also the work of Fiesel et al. 2014 (BMC Veterinary Research 10:196).
Author response: Thank you for this suggestion. We have now included the suggested references into the new version of the paper as follow:
“Studies in pigs fed grape pomace (GP) diet have shown that GP decrease the relative abundance of Lactobacillus and Ruminococcus [39]. Also, proanthocyanidins from grape seed decreased the abundance of Lactobacillaceae and increased the abundance of Clostridiaceae in both ileal and colonic lumen [40].”